# Experimental Studies of Scale Effect on the Shear Strength of Coarse-Grained Soil

**Shuya Li [1], Tiancheng Wang [1], Hao Wang [2], Mingjie Jiang [1,3,\*] and Jungao Zhu [4]**

1   College of Civil Engineering and Architecture, Guangxi University, Nanning 530004, China; lishuya@st.gxu.edu.cn (S.L.); wangtiancheng@st.gxu.edu.cn (T.W.)
2   Department of Civil and Environmental Engineering, Harbin Institute of Technology, Shenzhen 518055, China; wanghao20182021@163.com
3   Guangdong Key Laboratory of Oceanic Civil Engineering, Sun Yat-sen University, Zhuhai 510275, China
4   Key Laboratory of Ministry of Education for Geomechanics and Embankment Engineering, Hohai University, Nanjing 210024, China; zhujungao@hhu.edu.cn
\*   Correspondence: 20180121@gxu.edu.cn

**Abstract:** Shear strength is an essential index for the evaluation of soil stability. Test results of the shear strength of scaled coarse-grained soil (CGS for short) are usually not able to accurately reflect the actual properties and behaviors of in situ CGS due to the scale effect. Therefore, this study focuses on the influence of the scale effect on the shear strength of scaled CGS, which has an important theoretical significance and application for the strength estimation of CGS in high earth-rock dam engineering. According to previous studies, the main cause of the scale effect for scaled CGS is the variation of the gradation structure as well as the maximum particle size ($d_{max}$), in which the gradation structure as a characteristic parameter can be expressed by the gradation area (S). A total of 24 groups of test soil samples with different gradations were designed by changing the maximum particle size $d_{max}$ and gradation area S. Direct shear tests were conducted in this study to quantitatively explore the effect of the gradation structure and the maximum particle size on the shear strength of CGS. Test results suggest that the shear strength indexes (i.e., the cohesion and internal friction angle) of CGS present an increasing trend with the improvement of the maximum particle size $d_{max}$, and thus a logarithmic function relationship among $c$, $\varphi$, and $d_{max}$ is presented. Both cohesion ($c$) and internal friction angle ($\varphi$) are negatively related to the gradation area (S) in most cases. As a result, an empirical relationship between $c$, $\varphi$, and S is established based on the test results. Furthermore, a new prediction model of shear strength of CGS considering the scale effect is proposed, and the accuracy of this model is verified through the test results provided by relevant literature. Finally, the applicability of this model to different types of CGS is discussed.

**Keywords:** coarse-grained soil; scale effect; shear strength; maximum particle size; gradation structure

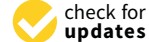



## 1. Introduction

In high-earth rock dam engineering, the mechanical characteristics of coarse-grained soil (CGS for short) play essential roles. Shear strength serves as an important mechanical index to evaluate the shear strength capacity of CGS in practical engineering. The accurate determination of the shear strength of soil (especially coarse-grained soil) is of great significance to the estimation of strength in high earth-rock dam engineering. Coarse-grained soil is utilized widely in high-earth rock dam projects; the maximum particle size of coarse-grained soil ranges from 800 to 1000 mm and may even reach 1200 mm in several special engineering projects [1]. Due to the dimension limitations of laboratory instruments, the original gradation of CGS needs to be scaled so as to determine its mechanical behavior [2]. However, no matter what adopted scale method is utilized, the inevitable scale effect is observed from test results [1,3]. This means that the test results for scaled CGS may not accurately reflect the actual properties of in situ CGS. As a result,

an in-depth study of the scale effect on the shear strength of scaled CGS is very important to understand the mechanical behaviors of CGS.

This study on the scale effect aims to (1) understand the difference of mechanical properties between scaled and in situ CGS and (2) accurately evaluate the mechanical properties of the prototype CGS [4]. Unfortunately, it is difficult to carry out reliable and accurate full-scale field tests using the existing test technology. At present, research on the scale effect on the mechanical behavior of CGS is mainly investigated through numerical simulations as well as laboratory experiments [3]. However, previous studies' results regarding the mechanical properties of CGS caused by the scale-induced effect usually deduce significant differences and even contrary conclusions [5,6]. Marsal [7] conducted a triaxial shear test on two groups of CGS with similar gradation and different maximum particle sizes. It was found that the larger the particle size, the lower the shear strength with the same dry density. Marachi et al. [8] used a similar gradation method and the same dry density sampling standard to carry out triaxial shear tests on CGS with different sizes. The results showed that the maximum particle size decreased as the maximum particle size increased. The shear test results were consistent with the findings of Varadarajan et al. [9]. This is because the difference among the soil sample preparation standards significantly affected the test results. Test results suggest that when soil samples of CGS have the same dry density, as the maximum particle size (i.e., $d_{max}$) increases, the shear strength of CGS also increases.

Nevertheless, there is still not a clear understanding of the scale-induced effect on shear strength when the sample preparation is conducted according to a standard with the same relative density [3]. For instance, Gupta et al. [10], Li et al. [11], and Kim et al. [12] support that the internal friction angle $\varphi$ increases with the improvement of $d_{max}$. On the contrary, Lee [13] and Hu et al. [14] believe that the maximum particle size $d_{max}$ has little effect on the peak internal friction angle $\varphi$. Xu et al. [15], Wei et al. [16], and Simoni et al. [17] found that the internal friction angle increases with the reduction of content of fine particles, and Li et al. [18] observed that when the fine particle content is relatively low, the internal friction angle is stable at a certain level; however, after that, it decreases rapidly with increasing fine-particle content.

Besides physical experiments, the discrete element method (DEM) was also a significant approach used to investigate the scale effect on the mechanical properties of CGS [19,20]. Since Cundall et al. [21] first applied the DEM in geotechnical engineering, many scholars use DEM software to explore the scale-induced effect in the shear strength of coarse-grained soil. For example, Frossad et al. [22] analyzed and summarized numerous test data according to the similarity theory of particle crushing strength, and deduced the strength envelope under various maximum particle sizes. Hence, they verified the existence of the influence of the scale effect on the mechanical properties of CGS from a theoretical point of view. Zhou et al. [23] investigated the impacts of particle shape on the mechanical properties of rockfill based on the discrete element method and observed that the shear strength increases with the improvement of the particle aspect ratio. Nimbkar [24] studied the influences of maximum particle size, particle gradation, and structure on the mechanical behaviors of granular materials through a discrete numerical method. Although the DEM method has advantages in exploring the meso mechanism of the scale-induced effect on mechanical characteristics and tracking the evolution law of meso fabric of CGS, numerical simulation results still need to be verified by laboratory tests.

Scaling methods for the gradation structure of CGS include similar grading methods, equivalent substitution methods, elimination methods, and mixing methods [2]. However, Guo et al. [25] found that no matter which scaling method is adopted, the gradation structure of CGS after scaling significantly varies from in situ CGS. In particular, the elimination method removes some ultra-fine particles of CGS, which may increase the content of fine particles. In contrast, the equivalent substitution method (ESM) maintains the coarse particle content of the original gradation structure, but the particle size range of the gradation structure becomes smaller and the uniformity increases after scaling.

The similar grading method (SGM) maintains the relative size between particles and the curvature coefficient of the non-uniformity coefficient. By analyzing the scaling methods, Wu et al. [26] found that SGM is used to reduce the graded particle size according to a certain proportion in the mixing method, and then ESM is used to scale samples. However, it is necessary to use the empirical index $P_5$ content to control the gradation structure of CGS ($P_5$ is the relativeness with the scale ratio or the maximum particle size of the scaled material). The mixing method is essentially an empirical method, and the accuracy of the test results is insufficient and unconvincing.

In conclusion, owing to the influence of scaling methods, gradation structure, sample preparation densities, and other factors, the relationship between the mechanical properties of scaled and in situ CGS is still difficult to describe quantitatively [4]. Most of the studies of scale effect on the shear strength consider the effect of maximum particle size and coarse-grain contents, which obviously cannot accurately represent the effect of scaled gradation on shear strength. Thus, it is necessary to conduct quantitative experimental investigations on the influence of the scale effect on the mechanical characteristics of CGS.

The main reason why it is difficult to quantitatively express the relationship between the mechanical characteristics of scaled and in situ soil is that there is no accurate and unified mathematical expression of gradation structure. Zhu et al. [27] sorted out and summarized the soil grading curve types in engineering projects, and proposed a new gradation equation suitable for CGS, as follows:

$$P = \frac{1}{(1-e)\left(\frac{d_{\max}}{d}\right)^n + b} \times 100\% \tag{1}$$

where $P$ denotes the content of soil particles that are smaller than particle size $d$ (%); $d$ represent the particle size of soil (mm), $d_{\max}$ is the maximum particle size of soil (mm); and $e$ and $n$ are the parameters of the gradation equation, which determine the shape and inclination of the grading curve, respectively (referred to as the gradation structure). The gradation parameters of coarse-grained soil in high-earth rock dam engineering are concentrated in the region of $-2 < e < 1$ and $0 < n < 2$, which includes both well-graded and poorly graded cases. The gradation of most coarse grains can be described by adjusting the combination of parameters $e$ and $n$. According to the relevant literature [26,28], regardless of which scaling method is adopted [1], the differences between the scaled and the in situ soil after scaling are mainly the gradation structure and the maximum particle size. As a result, the scale-induced effect can be regarded as the result of the coupling effect of the maximum particle size and gradation structure of the soil sample.

Therefore, in this paper, 24 groups of samples of CGS are designed by changing the maximum particle size or gradation structure. A series of large-direct shear tests on samples prepared with the same relative compactness is conducted to explore the relationship of the shear strength of scaled and in situ soil. Furthermore, the impact of gradation structure and maximum particle size upon the CGS is studied by a single variable method, and the relational equation of shear strength direct shear of scaled and in situ soil is established. Consequently, as long as a series of shear strength tests of laboratory scaled samples are carried out following the on-site graduation to acquire the material parameters for the equal soil types, CGS with any graduation can be predicted.

## 2. Materials and Methods

### 2.1. Test Apparatus

A large-scale direct-shear apparatus (Trac-III model) is used in the present study. The details of the apparatus are as shown in Figure 1, and the loading stress and structure diagram of the systems are given in Figure 2. The size of the shear box is $305 \times 305 \times 200$ mm (length $\times$ width $\times$ height) with a shear rate of 0.0003-15 mm/min. The maximum horizontal and vertical displacements are 75 mm, and the permissible horizontal and vertical loading is 50 kN. During the test, a vertical load with 1.0 kN was used to ensure the

component was in close contact status, and then to control the vertical pressure with 50, 100, 150, and 200 kPa, respectively. When the vertical deformation of soil sample is less than 0.03 mm/h, the consolidation can be considered to be completed. Accordingly, the maximum particle sizes were 40, 20, and 10 mm, respectively. Correspondingly, the spacing of the shear chink between the upper steel box and lower steel box is controlled at 12, 6, and 3 mm, respectively. Furthermore, the shear speed is controlled at 4 mm/min to start shearing process (meeting the requirements of the consolidated rapid shear test). The shearing process is stopped when the shearing displacement reaches 30 mm.

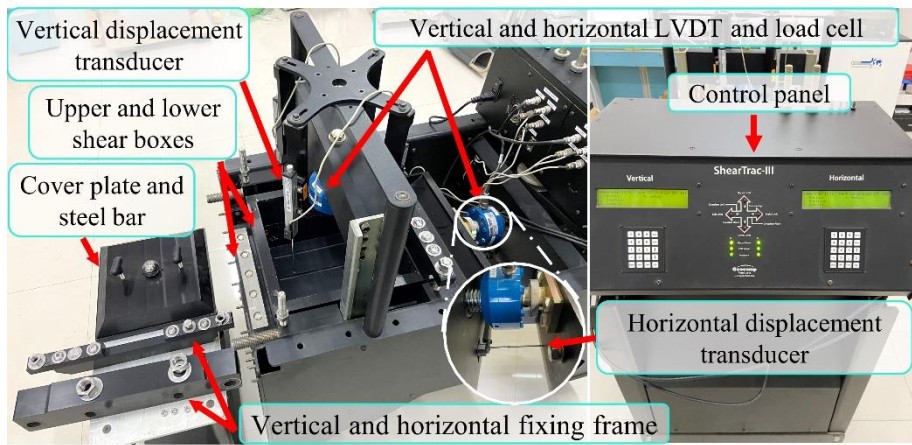

**Figure 1.** Large-scale direct shear system (Shear Trac III).

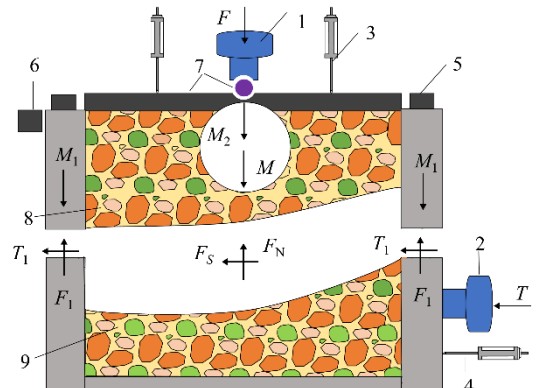

1—Vertical LVDT and load cell; 2—Horizontal LVDT and load cell;
3—Vertical displacement transducer; 4—Horizontal loading transducer;
5—Vertical fixing frame; 6—Horizontal fixing frame;
7—Cover plate and steel bar; 8—Upper shear box; 9—Lower shear box

**Figure 2.** Structure diagram and loading mode of the large-scale direct shear apparatus (Shear Trac III).

The shear stress $\tau$ and vertical pressure $\sigma$ can be obtained from the following Equations (2)–(5):

$$F_N = F + M + M_1 + M_2 - F_1 \tag{2}$$

$$F_S = T - T_1 \tag{3}$$

$$\tau = F_S / A \tag{4}$$

$$\sigma = F_N / A \tag{5}$$

where $M$ is the sample gravity of the upper shear box; $M_1$ is the gravity of the upper shear box; $M_2$ is the gravity of the cover plate and steel ball (taken as 100 N); $F$ is the applied vertical load; $T$ is the horizontal load of the shear box; and $F_1$ and $T_1$ are the vertical and horizontal load of the bottom surface of the upper shear frame and samples, respectively. When the soil sample reaches the peak shear strength during the direct shear process, it is

usually accompanied by shear expansion. In addition, the shear frame is in a suspended state due to the shear chink between the upper steel box and lower steel box ($F_1$ and $T_1$ are approximately 0, while $M_1$ can be ignored). The system during the test can automatically record $F_N$ and $F_S$, and then calculate the results of $\tau$ and $\sigma$.

### 2.2. Soil Samples

The soil samples are taken from sandy-grained soil, which was collected from a prototype site of an engineering project (refer to Figure 3). In order to avoid the negative influence of non-uniformity between different samples, it was necessary to classify the sandy-grained soil of the prototype site. The soil sample was then remixed according to the gradation structure of the scaled sandy-grained soil. The relative density $D_r$ of the sample was controlled at 0.80 and the water content was controlled at 5%. During sample preparation, the mass and water content of the sample required for each design group of samples was calculated according to the average air-dried moisture content (2.9%). Subsequently, the fresh-keeping film was sealed and stewed for 24 h after the mixing process to ensure the full penetration of water and soil. Finally, the weight of sand and pebbles required for the test was calculated according to the dry density and the size of the shear box during the filling process. After mixing the soil evenly, the soil (in three layers) was loaded into the shear box, roughened between layers, and rammed to the control height with a compaction hammer.

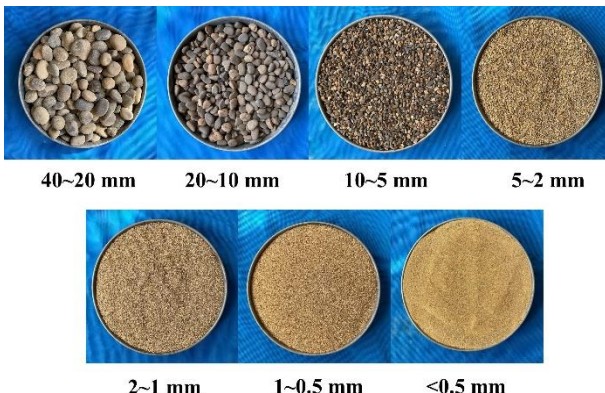

**Figure 3.** Sandy-gravel soil samples of different gradation groups.

A total of 24 groups of coarse-grained soil samples were investigated, of which the maximum particle sizes of the samples were 40, 20, and 10 mm, respectively. In addition, the particle size of the test soil was required to be less than the instrument size by about 1/5 to 1/6 (i.e., less than 50.8–61.0 mm). The details of the grading parameters and sample preparation density are listed in Tables 1 and 2. The parameter $S$ in Table 1 represents the place of the gradation area ($S$) as described in Section 2.3. Among them, there are 12 groups of samples with the same maximum particle size but different gradation structures, labeled A1-4 to A12-4, and the corresponding grading curve is shown in Figure 4a. Correspondingly, there are 12 groups of samples with the different maximum particle sizes and same gradation structures, labeled D1-4 to D4-4, and the corresponding grading curves are shown in Figure 4b.

**Table 1.** Details of grading parameters and grading area of soil samples.

| No. | $d_{max}$/mm | $n$ | $e$ | $S$ |
|---|---|---|---|---|
| A1-4 | 40 | 1.0 | 0.6 | 0.539 |
| A2-4 | 40 | 1.0 | −0.2 | 0.350 |
| A3-4 | 40 | 1.0 | −1.0 | 0.273 |
| A4-4 | 40 | 0.8 | 0.3 | 0.504 |
| A5-4 | 40 | 0.8 | −0.2 | 0.408 |
| A6-4 | 40 | 0.8 | −1.0 | 0.322 |

**Table 1.** *Cont.*

| No. | $d_{max}$/mm | $n$ | $e$ | $S$ |
|---|---|---|---|---|
| A7-4 | 40 | 0.6 | 0.6 | 0.673 |
| A8-4 | 40 | 0.6 | 0.3 | 0.581 |
| A9-4 | 40 | 0.6 | −0.2 | 0.482 |
| A10-4 | 40 | 0.4 | 0.6 | 0.749 |
| A11-4 | 40 | 0.4 | 0.3 | 0.672 |
| A12-4 | 40 | 0.4 | −1.0 | 0.486 |
| D1-4 | 40 | 1.0 | 0.3 | 0.441 |
| D1-2 | 20 | 1.0 | 0.3 | 0.441 |
| D1-1 | 10 | 1.0 | 0.3 | 0.441 |
| D2-4 | 40 | 0.8 | 0.6 | 0.603 |
| D2-2 | 20 | 0.8 | 0.6 | 0.603 |
| D2-1 | 10 | 0.8 | 0.6 | 0.603 |
| D3-4 | 40 | 0.6 | −1.0 | 0.390 |
| D3-2 | 20 | 0.6 | −1.0 | 0.390 |
| D3-1 | 10 | 0.6 | −1.0 | 0.390 |
| D4-4 | 40 | 0.4 | −0.2 | 0.581 |
| D4-2 | 20 | 0.4 | −0.2 | 0.581 |
| D4-1 | 10 | 0.4 | −0.2 | 0.581 |

**Table 2.** Test results of relative compactness of soil samples and dry density of soil samples.

| No. | $\rho_{min}$/(g·cm³) | $\rho_{min}$/(g·cm³) | $D_r$ | $\rho_0$/(g·cm³) |
|---|---|---|---|---|
| A1-4 | 1.905 | 2.163 | 0.8 | 2.106 |
| A2-4 | 1.815 | 1.994 | 0.8 | 1.955 |
| A3-4 | 1.750 | 1.959 | 0.8 | 1.913 |
| A4-4 | 1.878 | 2.122 | 0.8 | 2.068 |
| A5-4 | 1.838 | 2.049 | 0.8 | 2.003 |
| A6-4 | 1.782 | 1.961 | 0.8 | 1.922 |
| A7-4 | 1.877 | 2.171 | 0.8 | 2.105 |
| A8-4 | 1.909 | 2.197 | 0.8 | 2.133 |
| A9-4 | 1.894 | 2.127 | 0.8 | 2.076 |
| A10-4 | 1.846 | 2.139 | 0.8 | 2.073 |
| A11-4 | 1.899 | 2.199 | 0.8 | 2.132 |
| A12-4 | 1.897 | 2.149 | 0.8 | 2.093 |
| D1-4 | 1.833 | 2.106 | 0.8 | 2.045 |
| D1-2 | 1.774 | 2.051 | 0.8 | 1.989 |
| D1-1 | 1.641 | 1.962 | 0.8 | 1.888 |
| D2-4 | 2.086 | 2.426 | 0.8 | 2.349 |
| D2-2 | 1.908 | 2.252 | 0.8 | 2.174 |
| D2-1 | 1.741 | 2.115 | 0.8 | 2.028 |
| D3-4 | 1.906 | 2.155 | 0.8 | 2.100 |
| D3-2 | 1.830 | 2.104 | 0.8 | 2.043 |
| D3-1 | 1.664 | 2.002 | 0.8 | 1.924 |
| D4-4 | 1.973 | 2.302 | 0.8 | 2.228 |
| D4-2 | 1.842 | 2.199 | 0.8 | 2.117 |
| D4-1 | 1.688 | 2.028 | 0.8 | 1.949 |

*2.3. Research Method of Scale Effect*

In order to quantitatively explore the influence of the maximum particle size $d_{max}$ and gradation structure on the mechanical characteristic of CGS, the gradation equation of Equation (1) was selected to quantitatively express the CGS. Subsequently, the shear strength relationship of the maximum particle size $d_{max}$, gradation structure $n$, and gradation area $e$ were established. Through the analysis and summary of the test data, showing not direct relationship of $e$ and $n$ to $c$ and $\varphi$, a quantitative relationship of $e$ and $n$ with the grading area $S$ is presented. The grading area $S$ is surrounded by the grading equation curve, coordinate horizontal axis, maximum particle size $d = d_{max}$ line and $d = d_{max0}$ line (as given in Figure 5). Therefore, the calculation equation is as follows:

$$S = \frac{\ln(1 - we) - \ln(1 - e)}{ne \ln 10} \tag{6}$$

where $w$ is the percentage passing when $d_{\max} = d_{\max0}$, which can be expressed as:

$$w = \frac{1}{(1 - e)(d_{\max}/d_{\max0})^n + e} \tag{7}$$

According to Equations (6) and (7), a quantitative relationship between parameters $e$ and $n$ and the grading curve area is established. That is, the grading area value is affected by parameters $e$ and $n$.

According to the single variable methods [2], it should ensure that the grading curve area $S$ is the same value before and after scaling so as to reveal how the maximum particle size affects the shear strength of CGS. Based on the Equations (6) and (7), and the literature [25], the values of grading parameters $e$ and $n$ is maintained after the scaling of the similar grading method with a consistent $d_{\max}/d_{\max0}$, while the grading curve area $S$ is the same before and after scaling. The literature [2] also suggested the boundary particle size of sand and gravel as taken to be considered to be about 5 mm. Therefore, $d_{\max}$ and $d_{\max0}$ are taken as 40 and 5 mm, respectively, to solve the grading curve area $S$. At this time, the grading curve area $S$ is only related to the grading parameters $e$ and $n$, and the gradation structure is determined by both $e$ and $n$. As a result, the influence of the gradation structure on mechanical characteristics can be expressed by the grading area $S$ as an alternative characteristic parameter.

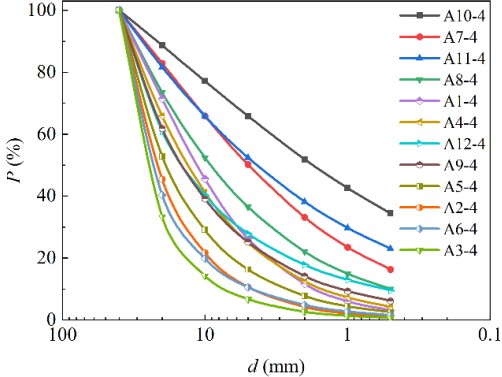

**(a)**

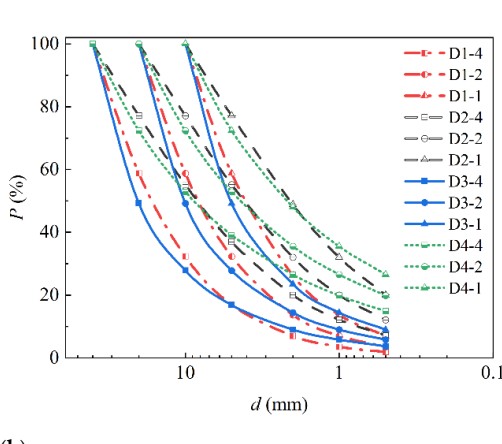

**(b)**

**Figure 4.** Gradation curves of test CGS. (**a**) Samples with same gradation structures and the different maximum particle sizes. (**b**) Samples with the different maximum particle sizes and same gradation structures.

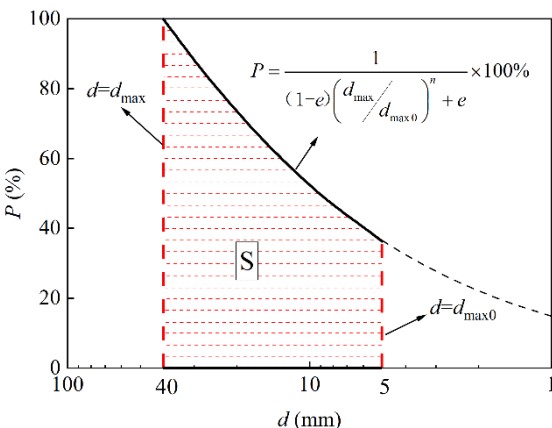

**Figure 5.** Schematic diagram of gradation curve area.

In summary, it is feasible that the control variable method is used to set a series of initial conditions of the same maximum particle size and different gradation structures, as well as the same gradation structure and different maximum particle sizes, to reveal the relationship of the shear strength of CGS with the maximum particle size and gradation structure.

## 3. Results

### 3.1. Relationship between Shear Strength Index and Maximum Particle Size

According to the Mohr–Coulomb theory [2,16], the shear strength of CGS can be expressed as Equation (8):

$$\tau = c + \sigma \tan \varphi \tag{8}$$

where $\sigma$ is the vertical pressure, $\tau$ denotes shear strength, $c$ is cohesion, and $\varphi$ represents the internal friction angle.

According to Equation (8), the shear strength of coarse-grained material is composed of cohesion $c$ and internal friction angle $\varphi$. In particular, for the coarse-grained soil composed of coarse and fine particles, the cohesion $c$ includes not only the interaction between coarse particles, but also the particle cohesion between fine particles.

A total of 24 groups of soil samples were tested; the vertical pressures of each group were 50, 100, 150 and 200 kPa, respectively. The relationship curve of shear strain ($\tau$) versus shear strain ($\gamma$) is drawn in Figure 6. The stress-strain curves of grade batching with $S = 0.581$ and $d_{max}$ values of 40, 20, and 10 mm are listed. The peak strength of shear stress is taken as the shear strength and fitted with Equation (8). The intercept of the straight line on the ordinate is presented as the cohesion $c$, and the inclination angle is presented as the internal friction angle $\varphi$ (refer to Figure 7).

According to the test results of D1-4 to D4-4, the maximum particle size $d_{max}$ and shear strength indexes are sorted out by cohesion $c$ and internal friction angle $\varphi$. The variation relationship is shown as the discrete points in Figures 8 and 9. As can be observed, the maximum particle size has a great impact on the shear strength characteristics of CGS. Furthermore, regardless of the gradation structure, both cohesion $c$ and internal friction angle $\varphi$ increase with the improvement of the maximum particle size $d_{max}$. The triaxial shear test results of rockfill materials for $c$ and $\varphi$ carried out by Li et al. [11] are consistent with those obtained in this test results; the change law is also similar to those of $d_{max}$. The ratio of coarse and fine particle content in coarse-grained soil is one essential factor affecting its shear strength [16]. From Equations (6) and (7), the grading area $S$ is proportional to the coarse content. The increase or decrease of gradation area $S$ will cause the degree of filling of coarse and fine particles to change, and the values of cohesion and internal friction angle will also change with it.

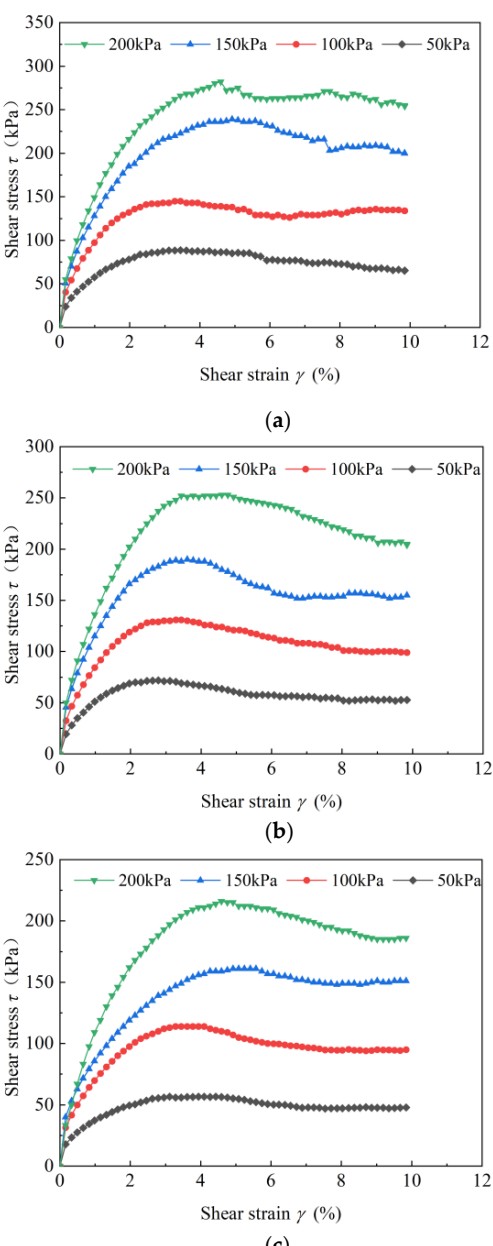

**Figure 6.** Relationship of shear stress and strain. (**a**) $S$ = 0.581, $d_{max}$ = 40. (**b**) $S$ = 0.581, $d_{max}$ = 20. (**c**) $S$ = 0.581, $d_{max}$ = 10.

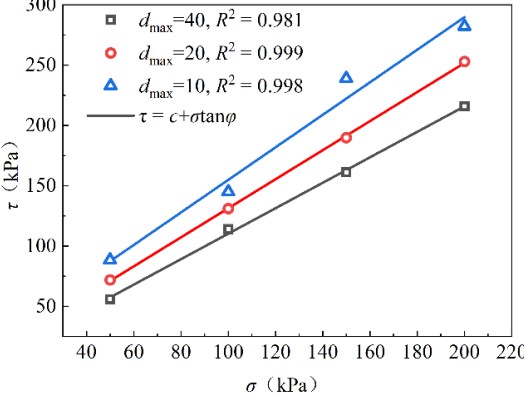

**Figure 7.** Relationship between internal axial pressure and shear strength of test CGS.

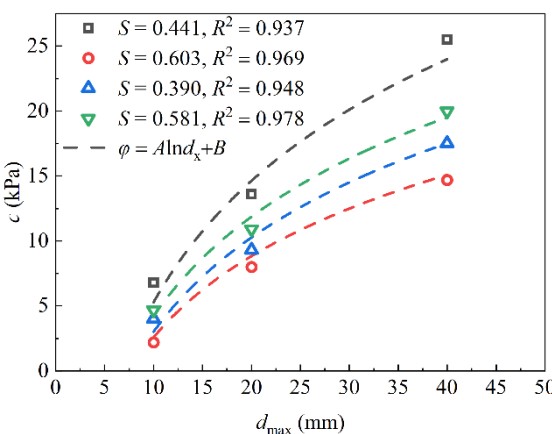

**Figure 8.** Relationship between internal maximum particle size and cohesion of test CGS.

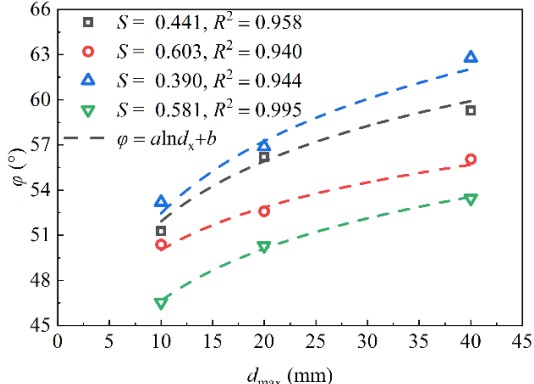

**Figure 9.** Relationship between maximum particle size and the internal friction angle of test CGS.

Further analysis found that the relationship of the $d_{max}$, $c$ and $\varphi$ can be expressed as a logarithmic equation. Hence, the relation of can be expressed as:

$$\begin{cases} c = a_1 \ln d_{max} + b_1 \\ \varphi = a_2 \ln d_{max} + b_2 \end{cases} \tag{9}$$

where $a_1$, $b_1$, $a_2$, and $b_2$ are parameters; $b_1$ and $b_2$ represent the cohesion and internal friction angle of the sample when the maximum particle size is 1 mm, and the units are kPa and (°), respectively; and $a_1$ and $a_2$ represent the change rate of cohesion and internal friction angle of the sample when the maximum particle size is 1 mm, and the units are kPa/ln(mm) and (°)/ln(mm), respectively.

The fitting curves according to Equation (9) are given in Figures 8 and 9. Furthermore, it is found that the fitting curves are in good agreement with the test data points, and the error between the fitting value and the corresponding test data points is basically less than 3% and 1%. The maximum error is less than 6.87% and 1.29%, and the determination coefficient is more than 0.937. Therefore, it implies that the impacts of the maximum particle size $d_{max}$ on internal friction angle $\varphi$ and cohesion $c$ in the scale effect can be described quantitatively by Equation (9).

### 3.2. Relationship of Shear Strength Index and Grading Area

The variation of internal friction angle $\varphi$ and cohesion $c$ with grading structure is shown in Figures 10 and 11. With the decrease of grading area $S$, the cohesion $c$ and internal friction angle $\varphi$ gradually increase from 11.1 to 32 kPa and 44.83° to 59.88°, respectively. The grading area $S$ continues to decrease, and the cohesion $c$ and internal friction angle $\varphi$ show a slightly decreasing trend. The differences between the maximum and minimum values

of the cohesion $c$ and internal friction angle $\varphi$ are 65% and 25%, respectively. Obviously, the gradation structure $S$ has a significant influence on the cohesion $c$ of the shear strength index of CGS. This may be because when the grading curve area $S$ is large or small, the filling degree of coarse and fine particles is low. When the grading curve area $S$ is about 0.4, the filling effect of coarse and fine particles is prospective, and the shear strength of soil material is large.

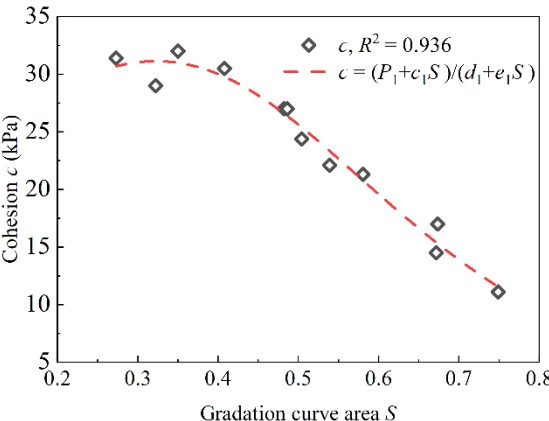

**Figure 10.** Relationship between gradation curve area and cohesion.

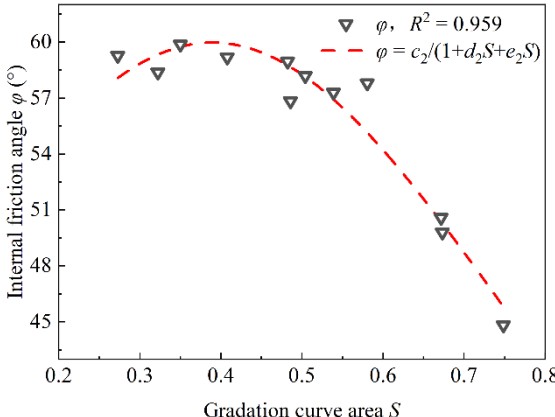

**Figure 11.** Relationship between gradation curve area and internal friction angle.

As a result, the relationship of the cohesion $c$, internal friction angle $\varphi$, and grading curve area $S$ can be expressed as:

$$\begin{cases} c = \frac{P_1 + c_1 S}{d_1 + e_1 S} \\ \varphi = \frac{c_2}{1 + d_2 S + e_2 S^2} \end{cases} \tag{10}$$

where $P_1$, $c_1$, $d_1$, $e_1$, $c_2$, $d_2$, and $e_2$ are all equation parameters, and the unit of $P_1$ and $c_1$ is kPa. In particular, $P_1$ is 1 kPa, the unit of $c_2$ is (°), and $d_1$, $d_2$, $e_1$, and $e_2$ are dimensionless.

The test data of soil samples A1-4 to A12-4 are fitted according to Equation (10), and the fitting curve is given in Figures 10 and 11. It is found that the fitting curve is in good agreement with respect to the test points. Most of the errors between the fitting value and the test value of $c$ and $\varphi$ are less than 5% and 2%, respectively. In particular, the maximum errors are less than 10.52% and 4.59%, respectively, and the determination coefficients are greater than 0.93. Therefore, it can be considered that the influence of scale effect on $c$ and $\varphi$ could be quantitatively described by Equation (10).

### 3.3. Derivation of Empirical Equation of Shear Strength of Coarse-Grained Soil

The relationship between cohesion $c$, internal friction angle $\varphi$, and grading area $S$ of CGS at a given maximum particle size satisfies Equation (9). It is mentioned that $b_1$ and $b_2$ in Equation (9) respectively represent the cohesion $c$ and internal friction angle $\varphi$ of the soil sample when the maximum particle size $d_{max}$ is 5 mm. Therefore, the relationship between $b_1$, $b_2$, and $S$ could be also expressed by Equation (10). That is, the influence of the maximum particle size $d_{max}$ and grading structure on the shear strength of CGS in the scale effect could be quantitatively described by Equation (11):

$$\begin{cases} c = a_1 \ln d_{max} + (P_1 + b_1 S)/(c_1 + d_1 S) \\ \varphi = a_2 \ln d_{max} + b_2/(1 + c_2 S + d_2 S^2) \end{cases} \tag{11}$$

where, $a_1$, $b_1$, $c_1$, $d_1$, $a_2$, $b_2$, $c_2$ and $d_2$ are taken as the equation parameters. $a_1$ and $a_2$, respectively, represent the change rate of the cohesion and the internal friction angle of the sample when the maximum particle size is 1 mm, and the units are kPa/ln(mm) and (°)/ln(mm), respectively. The unit of $P_1$ is kPa, where $P_1$ is 1 kPa. The units of $c_1$, $d_1$ and $d_2$ are dimensionless.

The shear strength index values of the 24 groups of the sand gravel test are fitted by using Equation (11), and the results are given in Table 3. The comparison between the measured values of the shear strength index of CGS in different gradation tests and the calculated values by Equation (11) are shown in Figures 12 and 13. The fitting value errors between $c$ and $\varphi$ are obtained by Equation (11), in which the cohesion $c$ and internal friction angle $\varphi$ of the corresponding test points are basically less than 7% and 3%, respectively, and the maximum errors are 10.85% and 5.81%, respectively. Notably, the determination coefficients are more than 0.865, which is in an acceptable range. Therefore, the scale-induced effect on shear strength of CGS could be quantitatively described by Equation (11).

In summary, using Equation (11) to reflect the variation law of the shear strength of CGS with respect to the maximum particle size and gradation structure has an important application. Based on the on-site grading of CGS, a series of laboratory-scaled material shear strength tests can be conducted to determine the material parameters through Equation (11) and to deduce the shear strength of the in situ graded CGS by combining Equation (8). This contributes to the improvement of the safety and reliability for the design of earth rock dam engineering.

**Table 3.** Fitting results of sandy-gravel soil.

| Fitting Parameters of Cohesion | | | | | Fitting Parameters of Internal Friction Angle | | | | |
|---|---|---|---|---|---|---|---|---|---|
| $a_1$ | $b_1/$(kPa) | $c_1$ | $d_1$ | $R^2$ | $a_2$ | $b_2/$(kPa·(°)$^{-1}$) | $c_2/$(kPa·(°)$^{-1}$) | $d_2$ | $R^2$ |
| 14.506 | −6.11 | 0.572 | −0.518 | 0.865 | 4.937 | 26.991 | −1.830 | 2.352 | 0.893 |

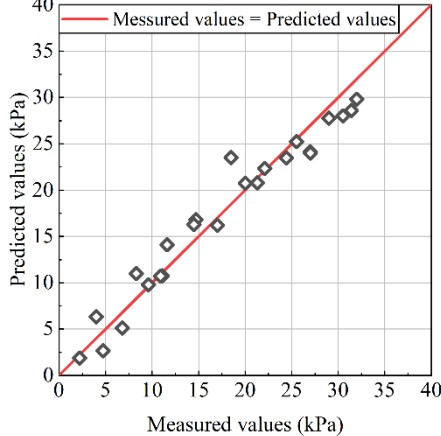

**Figure 12.** Measured and predicted values of the cohesion of CGS.

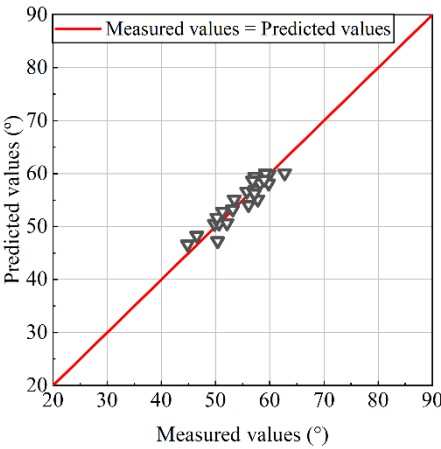

**Figure 13.** Measured and predicted values of the internal friction angle of CGS.

### *3.4. Verification of Empirical Equation of the Shear Strength of Coarse-Grained Soil*

In order to verify the applicability of the above shear strength prediction equation for different types of CGS, the above conclusions need to be supported by other test data. Wu et al. [29] carried out a series of large triaxial experiments on rockfill materials with different maximum particle sizes and grading structures by using the same fractal dimension grading design method and the same relative density sample preparation standard. It is mentioned that gneiss rockfill, as a medium-hard rock, is used in the test. The fractal dimensions $D$ is 2.3, 2.6, and 2.7, representing coarse batching, intermediate batching, and fine batching, respectively. The cohesion, $c$, internal friction angle, $\varphi$, and the grading structure parameter obtained by the optimal fitting of the in situ grading curve are listed in Table 4. The maximum particle sizes of the graded particles are 60, 40, and 20 mm, respectively. The maximum particle size grading forms of different particles remain similar and meet the grading characteristics after scaling by the similar grading method, as given in Figure 14.

It can be seen from Table 4 that when a sample preparation standard with the same relative density is adopted, the cohesion and internal friction angle increase with the increase of the maximum particle size, no matter what kind of gradation. This finding is consistent with the conclusion of these test results. However, compared with the sandy-grained soil, rockfill materials are less affected by the scale effect. The maximum and minimum values of the cohesion and internal friction angle of different dimensions are 5.17–17.31% and 3.16–10.33%, and the average values are 9.47% and 6.05%, respectively.

**Table 4.** Fitting results of test data of rockfill material soil.

| In Situ Dimension | Maximum Particle Size $d_{\max}$ | $n$ | $e$ | $S$ | Cohesion/kPa [29] | Internal Friction Angle/(°) [29] |
|---|---|---|---|---|---|---|
| | 60 | 0.7 | 0.003 | 0.512 | 185.808 | 39.68 |
| $D = 2.3$ | 40 | 0.7 | 0.003 | 0.512 | 177.727 | 39.4 |
| | 20 | 0.7 | 0.003 | 0.512 | 153.650 | 35.58 |
| | 60 | 0.4 | 0.02 | 0.686 | 204.094 | 41.41 |
| $D = 2.6$ | 40 | 0.4 | 0.02 | 0.686 | 200.382 | 41.14 |
| | 20 | 0.4 | 0.02 | 0.686 | 193.541 | 40.1 |
| | 60 | 0.3 | 0.09 | 0.779 | 209.852 | 41.72 |
| $D = 2.7$ | 40 | 0.3 | 0.09 | 0.779 | 197.401 | 39.78 |
| | 20 | 0.3 | 0.09 | 0.779 | 202.085 | 40.07 |

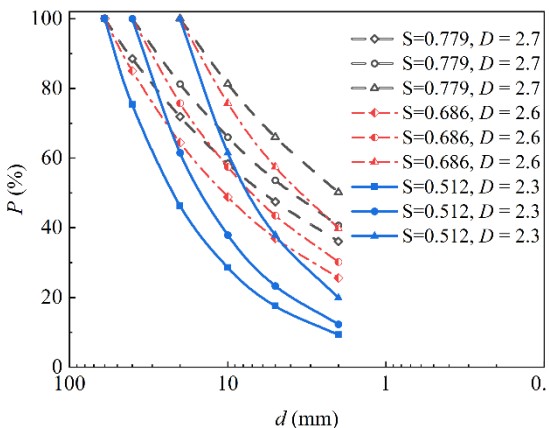

**Figure 14.** Gradation curves of rockfill materials [29].

The shear strength test results of the rockfill with $d_{max}$ of 40 mm and 20 mm are fitted by Equation (11), and the fitting parameters are listed in Table 5. The fitting values of $c$ and $\varphi$ are obtained by Equation (11), and the measured values of corresponding $c$ and $\varphi$ are plotted in Figures 15 and 16. It can be found that the $c$ and $\varphi$ are basically less than 2% and 1%, respectively, while the maximum errors are 4.81% and 3.79%, respectively. The determination coefficients are more than 0.867. It can be considered that the fitting empirical equation has a good applicability. The reliability of the prediction model and its applicability to different types of CGS are also explained. Therefore, as long as a series of shear strength tests of laboratory scaled samples are carried out according to the on-site gradation to obtain the material parameters of $c$ and $\varphi$, for the same soil types, the shear strength of CGS with different gradations can be predicted through Equations (8) and (11).

**Table 5.** Fitting results of the cohesion and internal friction angle of rockfill material soil.

| Fitting Parameters of Cohesion | | | | | Fitting Parameters of Internal Friction Angle | | | | |
|---|---|---|---|---|---|---|---|---|---|
| $a_1/(kPa{\cdot}lnmm^{-1})$ | $b_1/(kPa)$ | $c_1$ | $d_1$ | $R^2$ | $a_2/((°){\cdot}lnmm^{-1})$ | $b_2/(kPa{\cdot}(°)^{-1})$ | $c/(kPa{\cdot}(°)^{-1})$ | $d_2$ | $R^2$ |
| 14.778 | 1412.21 | 3.656 | 4.583 | 0.867 | 2.148 | 15.357 | −1.520 | 1.075 | 0.893 |

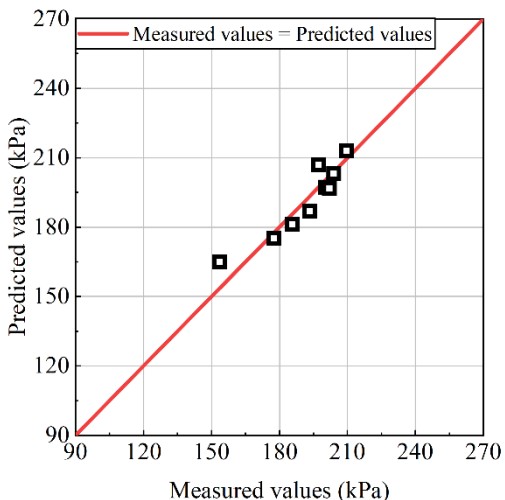

**Figure 15.** Measured and predicted values of the cohesion angle of rockfill materials.

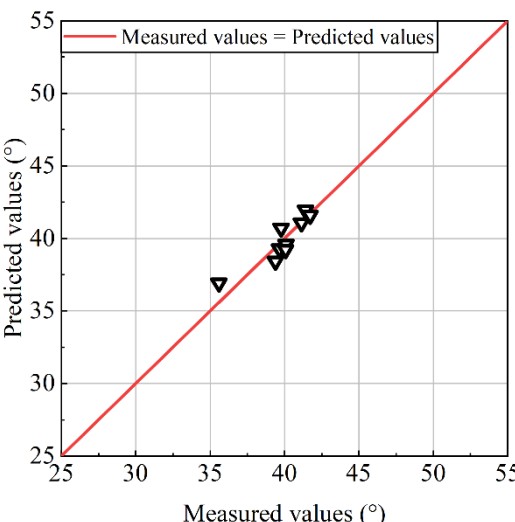

**Figure 16.** Measured and predicted values of the internal friction angle of rockfill materials.

## 4. Conclusions

Based on the continuous grading equation of soil and the idea of scale reduction via the similar grading method, 24 groups of test gradations with a maximum particle size $d_{max}$ of 40, 20, and 10 mm, respectively, were utilized to quantitatively investigate the shear strength of CGS with different gradation structures and maximum particle sizes under the same relative density by a large-scale direct shear instrument. The main conclusions obtained in the present research are as follows:

(1) Based on the continuous grading equation of soil, combined with the similar grading method, the purpose of quantitatively and comprehensively studying the scale-induced effect on the shear strength of CGS under consideration of the maximum particle size and gradation structure can be achieved;

(2) When the grading area is fixed such that the gradation structure is unchanged, the cohesion $c$ and internal friction angle $\varphi$ gradually increase with increases in the maximum particle size $d_{max}$; furthermore, $c$ and $\varphi$ are logarithmic functions of $d_{max}$;

(3) If the maximum particle size $d_{max}$ is constant, the cohesion and internal friction angle increase rapidly with the reduction of grading area $S$ and decrease slightly after reaching a certain level. The relationships between $c$, $\varphi$, and grading area $S$ are established;

(4) A new prediction model of the shear strength of CGS considering the influence of the maximum particle size $d_{max}$ and the gradation structure is established. The reliability of the model to predict the shear strength of arbitrarily graded CGS is verified through test data in the relevant literature, and the applicability of the model to different types of CGS is also explored.

**Author Contributions:** Conceptualization, S.L., T.W. and M.J.; methodology, M.J.; validation, J.Z.; formal analysis, S.L. and T.W.; investigation, M.J.; resources, J.Z.; data curation, S.L., T.W. and H.W.; writing—original draft, S.L., writing—review and editing, T.W., H.W. and M.J.; supervision, J.Z.; project administration, M.J.; funding acquisition, M.J. and J.Z. All authors have read and agreed to the published version of the manuscript.

**Funding:** This work was supported by the Yalong River Joint Fund of Natural Science Foundation of China (U1865104), key project of the Yangtze River Water Science Research Joint Fund of National Natural Science Foundation of China (U2040221) and the Foundation of Guangdong Key Laboratory of Oceanic Civil Engineering (LMCE202103).

**Institutional Review Board Statement:** Not applicable.

**Informed Consent Statement:** Not applicable.

**Data Availability Statement:** Not applicable.

**Acknowledgments:** This study was supported by the College of Civil Engineering and Architecture in Guangxi University.

**Conflicts of Interest:** The authors declare no conflict of interest.

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
