# Peer review of "Experimental Studies of Scale Effect on the Shear Strength of Coarse-Grained Soil"

_applsci, doi:10.3390/app12010447_

Round 1

Reviewer 1 Report

Dear Authors,

     I had the opportunity to review the paper proposed for Applied Sciences under the name "Experimental Studies of Scale Effect on the Shear Strength of Coarse-grained Soil". The study focuses on the influence of the scale effect on shear strength of scaled CGS (the scaled coarse-grained soil). The manuscript looks elaborate and brings interesting finding. Despite interesting topic and analyses, I present some suggestions which should be considered in my opinion or add. Comments are set out in the attachment.

Author Response

Dear Reviewers

Thank you for your letter and the comments on our paper entitled “Experimental Studies of Scale Effect on the Shear Strength of Coarse-grained Soil”. The reviewers’ comments are all valuable to our research and very helpful to improve the quality of our manuscript. The authors revised the manuscript carefully according to the comments and made necessary corrections. We sincerely hope that the revised manuscript will meet the requirement of your journal for potential publication. The detailed responses to the reviewers’ comments are presented as follows.

Thank you and best regards.

Yours sincerely,

Shuya Li, College of Civil Engineering and Architecture, Guangxi University, Nanning 530004, China.

Email: lishuya@st.gxu.edu.cn

Mingjie Jiang, College of Civil Engineering and Architecture, Guangxi University, Nanning 530004, China.

Email: 20180121@gxu.edu.cn

Comment 1:

The introduction should present define the purpose of the work and its significance, including specific hypotheses being tested. In case point out controversial and diverging hypotheses when necessary. Each reference should be referred to at length. The authors cite references in whole sets, e.g. line 62, 73, 89 etc.

Response 1:

Thanks for the reviewer’s suggestion. The cited references have been described in more detail. Added in Line 152,187 and 197:

 “Marsal [7] conducted a triaxial shear test on two groups of CGS with similar gradation and different maximum particle sizes. It was found that the larger the particle size, the lower the shear strength with a same dry density. Marachi et al. [8] used a similar gradation method and same dry density sampling standard to carry out triaxial shear tests on CGS with different sizes, the results showed that the maximum particle size decreased as the maximum particle size increased. The shear test results were consistent with Varadarajan et al. [9] findings.”

However, Guo et al. [25] found that no matter which scaling method is adopted, the gradation structure of CGS after scaling behavior significantly varies from those of in-situ CGS.

By analyzing the scaling methods, Wu et al. [28] found that SGM is used to reduce the graded particle size according to a certain proportion in the mixing method,

Comment 2:

All manuscripts must contain the required sections: Author Information, Abstract, Keywords, Introduction, Materials and Methods, Results, Conclusions etc. The authors chose to name the selected chapters differently, for example Test design (instead of Material and Method) or Test Results (instead of Results). I recommend changed names selected of chapters. Lines 128 and 209.

Response 2:

Thanks for the comments from the reviewer. We quite agree with you point of view. The whole manuscript has been carefully checked.

Comment 3:

In line 300. I think has the wrong classification. I suppose this should be correctly rewritten from chapter 4 to subhead 3.4. Verification of empirical equation of the shear strength of coarse-grained soil.

Response 3:

Thanks for the comments from the reviewer. We quite agree with you point of view. The chapter of verification of empirical equation of the shear strength of coarse-grained soil has been arranged from chapter 4 to 3.4. Many thanks for the reviewer’s efforts on our manuscript.

Comment 4:

All manuscripts must contain the required sections: Funding Information, Author Contributions, Conflict of Interest and other Ethics Statements. Authors don't give to mention the mention parts. Please, add missing parts.

Response 4:

Thanks for the comments from the reviewer. The Funding Information, Author Contributions, Conflict of Interest and other Ethics Statements has been given at the end of this paper.

Comment 5:

In line 188. Equation description of variables is missing 6 a 7. Please, add missing description.

Response 5:

Thanks for the comments from the reviewer. Equation of variables has been described now.

where w is the percentage passing when dmax = dmax0, which can be expressed as:

                                                  (7)

Comment 6:

Lines 232, 302 contain incorrect citations of used literature.

Response 6:

The cited references have been rewritten in the revised manuscript according to the required format.

Comment 7:

Authors stated extensive used references (47) but in the part the results, hardly not exist none at all the discussion. Please, add missing discussion.

Response 7:

Thanks for the comments from the reviewer. We quite agree with you point of view. The used references [29] has been discussed and compared with this study now.

It can be seen from Table 4 that when the sample preparation standard with a same relative density is adopted, the cohesion and internal friction angle increase with the increase of the maximum particle size, no matter what kind of gradation, which is con-sistent with the conclusion of this test results. However, compared with the sandy-grained soil, rockfill materials is less affected by the scale effect. The maximum and minimum values of cohesion and internal friction angle of different dimensions are 5.17 %–17.31 % and 3.16 %–10.33 %, and the average values are 9.47 % and 6.05 %.

  1. Wu, L.Q.; Ye, F.; Li W.Q. Experimental study on scale effect of mechanical properties of rockfill materials. Chin. J. Geotech. Eng. 2020, 42, 141-145

Reviewer 2 Report

No figures and tables in the present manuscript. Please resubmit your work.

Author Response

Dear Reviewers

Thank you for your letter and the comments on our paper entitled “Experimental Studies of Scale Effect on the Shear Strength of Coarse-grained Soil”. The reviewers’ comments are all valuable to our research and very helpful to improve the quality of our manuscript. The authors revised the manuscript carefully according to the comments and made necessary corrections. We sincerely hope that the revised manuscript will meet the requirement of your journal for potential publication. The detailed responses to the reviewers’ comments are presented as follows.

Thank you and best regards.

Yours sincerely,

Shuya Li, College of Civil Engineering and Architecture, Guangxi University, Nanning 530004, China.

Email: lishuya@st.gxu.edu.cn

Mingjie Jiang, College of Civil Engineering and Architecture, Guangxi University, Nanning 530004, China.

Email: 20180121@gxu.edu.cn

Comment 1:

No figures and tables in the present manuscript. Please resubmit your work.

Response 1:

Thanks for the comments from the reviewer. We quite agree with you point of view and we are very sorry for the trouble caused to you due to our unexpected mistakes in edit work. All figures and Tables have been added in the revised manuscript. Many thanks for the reviewer’s efforts on our manuscript.

Reviewer 3 Report

Dear Authors,

I could not review the manuscript because you did attach neither tables, neither graphs, which are necessary to describe experimental parts and  results. I read the text. The text seems to be tidy, although neither not very novel nor very interesting. It looks like some general description. Surely, the title and all their parts should be much more detailled. The material should be described, etc. 

In such situation I should reject the paper.

However, I encourage you  to thoroughly improve and supplement the manuscript and resubmit again.

Sincerely yours,

Reviewer

Author Response

Dear Reviewers

Thank you for your letter and the comments on our paper entitled “Experimental Studies of Scale Effect on the Shear Strength of Coarse-grained Soil”. The reviewers’ comments are all valuable to our research and very helpful to improve the quality of our manuscript. The authors revised the manuscript carefully according to the comments and made necessary corrections. We sincerely hope that the revised manuscript will meet the requirement of your journal for potential publication. The detailed responses to the reviewers’ comments are presented as follows.

Thank you and best regards.

Yours sincerely,

Shuya Li, College of Civil Engineering and Architecture, Guangxi University, Nanning 530004, China.

Email: lishuya@st.gxu.edu.cn

Mingjie Jiang, College of Civil Engineering and Architecture, Guangxi University, Nanning 530004, China.

Email: 20180121@gxu.edu.cn

Comment 1:

No figures and tables in the present manuscript. Please resubmit your work.

Response 1:

Thanks for the comments from the reviewer. We quite agree with your point of view and we are very sorry for the trouble caused to you due to our unexpected mistakes in edit work. All figures and Tables have been added in the revised manuscript. Many thanks for the reviewer’s efforts on our manuscript.

Round 2

Reviewer 2 Report

The manuscript presents an experimental study of the scale effect on the shear strength of CGS. The overall writing and novelty of this work is basically accepted. The discussions on experimental results can be deepen and some incorrections should be revised. As a result, my suggestion is to reconsider after revision.

  1. English language of the manuscript can be improved.
  2. Potential applications of this work should be discussed in the introduction.
  3. The novelty of this study should be highlighted in the introduction.
  4. In this study, the gradation structure is a significant factor determined by the shape index (e) and inclination index (n) of grading curve. I suggest the authors can discuss these two parameters in detail.
  5. In Table 1 and Table 2, there are two D4-4 in the same table.
  6. In Table 1, what is the definition of S.
  7. Line 226, the same maximum particle size?
  8. In Line 280, ΔL is normally the symbol for differential displacement rather than strain.
  9. The caption of figure 6, relationship of shear stress and strain?
  10. In Figure 8-9, the effect of S should be explained. I observed the variation law of c and φ with S are different.
  11. In section 3.2, the maximum particle size is 40mm?
  12. In the proposed fitting equations, do the fitting parameters have physical meaning?
  13. It is suggested to add some quantitative conclusions.

Author Response

Dear Reviewers

Thank you for your letter and the comments on our paper entitled “Experimental Studies of Scale Effect on the Shear Strength of Coarse-grained Soil”. The reviewers’ comments are all valuable to our research and very helpful to improve the quality of our manuscript. The authors revised the manuscript carefully according to the comments and made necessary corrections. We sincerely hope that the revised manuscript will meet the requirement of your journal for potential publication. The detailed responses to the reviewers’ comments are presented as follows.

Thank you and best regards.

Yours sincerely,

Shuya Li, College of Civil Engineering and Architecture, Guangxi University, Nanning 530004, China.

Email: lishuya@st.gxu.edu.cn

Mingjie Jiang, College of Civil Engineering and Architecture, Guangxi University, Nanning 530004, China.

Email: 20180121@gxu.edu.cn

Comment 1:

English language of the manuscript can be improved.

Response 1:

Thanks for the reviewer’s suggestion. I have invited a native English-speaking colleague to check the manuscript and will use the editing services recommended by the website if further touch-ups are needed before publication

Comment 2:

Potential applications of this work should be discussed in the introduction.

Response 2:

Thanks for the comments from the reviewer. The potential applications have been described in more detail. Added in Line 302.

Furthermore, the impact of gradation structure and maximum particle size upon the CGS is studied by a single variable method, and the relational equation of shear strength direct shear of scaled and on-situ soil is established. Consequently, as long as a series of shear strength tests of laboratory scaled samples are carried out following the on-site graduation to acquire the material parameters for the equal soil types, the CGS with any graduation can be predicted.

Comment 3:

The novelty of this study should be highlighted in the introduction.

Response 3:

Thanks for the comments from the reviewer. We quite agree with your point of view. More novelty of this study was added to the introduction. Added in Line 273.

In conclusion, owing to the influence of scaling methods, gradation structure, sample preparation densities, and other factors, the relationship between the mechanical properties of scaled and in-situ soil of CGS is still difficult to be described quantitatively [4]. Most of the studies of scale effect on the shear strength consider the effect of maximum particle size and coarse-grain contents, which obviously cannot accurately represent the effect of scaled gradation on shear strength. Thus, it is necessary to conduct quantitative experimental investigations on the influence of scale effect on the mechanical characteristic of CGS.

Comment 4:

In this study, the gradation structure is a significant factor determined by the shape index (e) and inclination index (n) of grading curve. I suggest the authors can discuss these two parameters in detail.

Response 4:

Thanks for the comments from the reviewer. The relevant information has been added to the paper in line 289.

The gradation parameters of coarse-grained soil in the high-earth rock dam engineering are concentrated in the region of -2<e<1 and 0<n<2, which includes both well-graded and poorly graded cases. The gradation of most coarse grains can be described by adjusting the combination of parameters e and n.

Comment 5:

In Table 1 and Table 2, there are two D4-4 in the same table.

Response 5:

Thanks for the comments from the reviewer. We quite agree with your point of view. The whole manuscript has been carefully checked.

Comment 6:

In Table 1, what is the definition of S.

Response 6:

Thanks for the comments from the reviewer. The parameter S in Table 1 represents the place of the gradation area (S), as described in Section 2.3. The relevant information has been added to the paper in line 362.

The parameter S in Table 1 represents the place of the gradation area (S), as described in Section 2.3.

Comment 7:

Line 226, the same maximum particle size?

Response 7:

Thanks for the comments from the reviewer. We quite agree with you point of view. The whole manuscript has been carefully checked.

Comment 8:

In Line 280, ΔL is normally the symbol for differential displacement rather than strain.

Response 8:

Thanks for the comments from the reviewer. We quite agree with you point of view and have revised it in my paper.

Comment 9:

The caption of figure 6, relationship of shear stress and strain?

Response 9:

Thanks for the comments from the reviewer. We quite agree with you point of view and have revised it in my paper.

Comment 10:

In Figure 8-9, the effect of S should be explained. I observed the varia tion law of c and φ with S are different.

Response 10:

Thanks for the comments from the reviewer. The relevant information has been added to the paper in line 442.

The ratio of coarse and fine particle content in coarse-grained soil is one of the essential factors affecting its shear strength [16]. From equations (6) and (7), the grading area S is proportional to the coarse content. The increase or decrease of gradation area S will cause the degree of filling of coarse and fine particles to change, and the values of cohesion and internal friction angle will also change with it.

Comment 11:

In section 3.2, the maximum particle size is 40mm?

Response 11:

Thanks for the comments from the reviewer. In this section, the main study is the relationship between shear strength index and gradation area, so the maximum particle size must be kept consistent

Comment 12:

In the proposed fitting equations, do the fitting parameters have physical meaning?

Response 12:

Thanks for the comments from the reviewer. In this paper, some of the equation parameters have physical meaning and units, and some of them are dimensionless. Example 1, in line 291, equation (9)

 where, a1, b1, a2, and b2 are parameters. b1 and b2 represent the cohesion and internal friction angle of the sample when the maximum particle size is 1 mm, and the unit are kPa and (º), respectively; a1 and a2 represent the change rate of cohesion and internal friction angle of the sample when the maximum particle size is 1 mm, and the unit are kPa/ln(mm) and (º)/ln(mm), respectively.

Example 2, in line 338, equation (11)

where, a1, b1, c1 and d1; a2, b2, c2 and d2 are taken as the equation parameters. a1 and a2 respectively represent the change rate of cohesion and internal friction angle of the sample when the maximum particle size is 1 mm, and the unit are kPa/ln(mm) and (º)/ln(mm), respectively; The unit of P1 is kPa, where P1 takes 1kPa. The unit of c1, d1 and d2 are dimensionless.

Comment 13:

It is suggested to add some quantitative conclusions

Response 13:

Thanks for the comments from the reviewer. The shear strength prediction model in conclusion (4) of this paper can quantitatively predict the shear strength of the original graded coarse-grained soil in the field. When used in the actual project, it is only necessary to do a series of shear strength tests of laboratory scaled samples are carried out according to the on-site graduation to obtain the material parameters of ϲ and φ, for the same soil types, the CGS with different gradation can be predicted through Equations (11) and (8).

Reviewer 3 Report

Dear Authors,

thank you for improving the manuscript to the state I could accept it.

Sincerely yours,

Reviewer

Round 3

Reviewer 2 Report

The new version has been properly revised based on the review comments. Please check the manuscript carely again before publication.

In Figure 8 and Figure 9, the fitting equations are not the same as Euqation 9. Please correct the mistake.